



# Indicators of Antarctic ozone depletion: 1979 to 2019

Greg E. Bodeker[1,2] and Stefanie Kremser[1]

[1]Bodeker Scientific, 42 Russell Street, Alexandra, 9320, New Zealand
[2]School of Geography, Environment and Earth Sciences, Victoria University of Wellington, New Zealand

**Correspondence:** G. E. Bodeker (greg@bodekerscientific.com)

**Abstract.** The National Institute of Water and Atmospheric Research/Bodeker Scientific (NIWA–BS) total column ozone (TCO) database, and the associated BS–filled TCO database, have been updated to cover the period 1979 to 2019, bringing both to version 3.5.1 (V3.5.1). The BS–filled database builds on the NIWA–BS database by using a machine-learning algorithm to fill spatial and temporal data gaps to provide gap-free TCO fields over Antarctic. These filled TCO fields then provide a more complete picture of winter-time changes in the ozone layer over Antarctica. The BS–filled database has been used to calculate continuous, homogeneous time series of indicators of Antarctic ozone depletion from 1979 to 2019, including (i) daily values of the ozone mass deficit based on TCO below a 220 DU threshold, (ii) daily measures of the area over Antarctica where TCO levels are below 150 DU, below 220 DU, more than 30 % below 1979 to 1981 climatological means, and more than 50 % below 1979 to 1981 climatological means, (iii) the date of disappearance of 150 DU TCO values, 220 DU TCO values, values 30 % or more below 1979 to 1981 climatological means, and values 50 % or more below 1979 to 1981 climatological means, for each year, and (iv) daily minimum TCO values over the range 75°S to 90°S equivalent latitude. Since both the NIWA–BS and BS–filled databases provide uncertainties on every TCO value, the Antarctic ozone depletion metrics are provided, for the first time, with fully traceable uncertainties. To gain insight into how the vertical distribution of ozone over Antarctica has changed over the past 36 years, ozone concentrations, combined and homogenized from several satellite-based ozone monitoring instruments as well as the global ozonesonde network, were also analysed. A robust attribution to changes in the drivers of long-term secular variability in these metrics has not been performed in this analysis. As a result, statements about the recovery of Antarctic TCO from the effects of ozone depleting substances cannot be made. That said, there are clear indications of a change in trend in many of the metrics reported on here around the turn of the century, close to when Antarctic stratospheric concentrations of chlorine and bromine peaked.

## 1 Introduction

The Antarctic ozone hole, discovered in the mid-1980s (Farman et al., 1985), is perhaps the best known manifestation of the impacts of ozone depleting substances (ODSs) on the global ozone layer. The primary role of ODSs as the cause of Antarctic ozone destruction has been firmly established on a body of evidence including laboratory measurements, atmospheric observa-



tions, and modeling studies (Newman et al., 2009). The Montreal Protocol, enacted in 1987, with subsequent Amendments and
Adjustments, committed countries to significantly reduce their production of ODSs. The Protocol has been labelled as one of
the most successful global environmental treaties (Gonzalez et al., 2015; McKenzie et al., 2019). The reduction in stratospheric
chlorine and bromine loading resulting from compliance with the Protocol has led to a recovery of ozone from the effects of
ODSs in many regions of the atmosphere (e.g. Yang et al., 2008; Kuttippurath, 2013; Solomon et al., 2016). Nevertheless, an
ozone hole continues to appear over Antarctica in each austral spring (Douglass et al., 2011). Projections of the future evolution
of the ozone layer over Antarctica using coupled chemistry-climate models suggest that with continued compliance with the
Protocol, the ozone layer over Antarctica is expected to return to 1980 levels around 2060 (Dhomse et al., 2018; Amos et al.,
2020).

Three metrics commonly used to define the Antarctic ozone hole are the area of the hole (adding the areas of cells falling
below some threshold in a TCO field), the minimum TCO value within the hole, and the Antarctic ozone mass deficit (Uchino
et al., 1999; Huck et al., 2007). Bodeker et al. (2005) reported on these metrics, using four different criteria for ozone hole type
values, viz. (i) TCO below 150 DU, (ii) TCO below 220 DU, (iii) TCO 30 % or more below the 1979 to 1981 climatological
mean, and (iv) TCO 50 % or more below the 1979 to 1981 climatological mean. Time series of these metrics were were
updated in subsequent publications (Müller et al., 2008; Struthers et al., 2009). Other studies to date have shown that all three
metrics show a slowing of Antarctic ozone depletion, consistent with the first stage of ozone recovery from the effects of ODSs
(Krzyścin et al., 2005; Keeble et al., 2018) around the turn of the century when Equivalent Effective Antarctic Stratospheric
Chlorine (EEASC; Newman et al., 2006) maximized.

Interannual variability in Antarctic stratospheric dynamics, manifest most obviously in interannual variability in Antarctic
stratospheric temperatures, drives significant interannual variability in the severity of Antarctic ozone depletion (Schoeberl
et al., 1996; Newman and Nash , 2000; Newman et al., 2004, 2006). Using TCO measurements from multiple satellite-based
instruments, and after accounting for interannual variability in stratospheric temperatures, de Laat et al. (2017) found that ozone
mass deficit decreased by $0.77 \pm 0.17$ Mt yr$^{-1}$ ($2\sigma$) between 2000 and 2015. More recently, Tully et al. (2019) analyzed linear
trends in several Antarctic ozone hole metrics over the periods 1979 to 2001 and 2001 to 2017. They considered metrics both
with and without an adjustment to account for interannual meteorological variability and found that all metrics they considered
showed a trend towards reduced ozone depletion since 2001 at significance levels between 2.4 and 3.9 standard errors of the
trend. However, they only used TCO measurements from three satellite instruments, i.e. TOMS, OMI and OMPS. Furthermore,
both de Laat et al. (2017) and Tully et al. (2019) appear not to have homogenized the different data sets used and did not infer
missing measurements during the polar night and elsewhere.

This study presents updated time series of metrics of the Antarctic ozone hole calculated from the long-term homogenised
National Institute of Water and Atmospheric Research/Bodeker Scientific (NIWA–BS) TCO database and the associated
Bodeker Scientific filled (BS–filled) TCO database, described in Sec. 2. It adds to a continuous body of literature reporting on
the evolution of the Antarctic ozone hole.



## 2 Ozone databases

This paper takes advantage of several features of the new version 3.4 (V3.4) NIWA–BS and BS–filled TCO databases (Bodeker
et al., 2020a), updates them to the end of 2019 to create V3.5.1 of the databases, and uses the BS–filled database to define
continuous, homogeneous time series of several metrics describing key attributes of the Antarctic ozone hole. The databases are
constructed using measurements from 17 different satellite-based instruments wherein offsets and drifts between (i) the ground-
based Dobson and Brewer spectrophotometer networks and (ii) a subset of the satellite-based measurements, are removed and
then used as the basis for homogenising the remaining TCO data sets. V3.4 and V3.5.1 of the BS–filled TCO databases
comprise spatially filled TCO fields that use a machine-learning approach to infer missing data in regions and at times for
which measurements were not available (Bodeker et al., 2020a). This approach significantly improves on the 'over the pole'
method described in Bodeker et al. (2001a) to create far more physically plausible renditions of the ozone fields in regions of
missing data. The result is a continuous gap-free database of daily TCO fields at $1.25°$ longitude by $1°$ latitude resolution.

Unlike previous versions of the database, both V3.4 (1979–2016) and V3.5.1 (1979–2019) include uncertainties traceable to
uncertainties in the TCO fields measured by the 17 different space-based instruments that are used to construct the database. To
propagate the uncertainties on the TCO fields to uncertainties in the Antarctic ozone depletion metrics, two additional databases
were created, one in which the $1\sigma$ uncertainties were added to each TCO field, and another where the $1\sigma$ uncertainties were
subtracted from each TCO field, i.e. assuming uniform over-estimation of TCO and under-estimation of ozone by $\pm 1\sigma$. By
calculating the ozone depletion metrics across all three databases, estimates can be obtained for $+1\sigma$ and $-1\sigma$ uncertainties on
all metric time series.

To demonstrate how the vertical structure of the ozone layer over Antarctica has changed from 1985 to 2019, ozone con-
centrations were extracted from the Bodeker Scientific vertically resolved ozone database (BSVertOzone, Hassler et al., 2018),
and mapped onto an equivalent latitude ($\phi_{eq}$) coordinate system so that only values well inside the ozone hole ($\phi_{eq}$ poleward
of $75°$S) could be selected. BSVertOzone combines measurements from several satellite-based instruments and ozone profile
measurements from the global ozonesonde network to create sparse fields of ozone concentrations on 70 altitude levels from 1
to 70 km. Offsets and drifts between each satellite-based ozone data set and a selected standard (SAGE-II in the stratosphere
and ozonesondes in the troposphere) was used to create a single homogeneous database of ozone concentrations. Similar to
the TCO database, measurement uncertainties and uncertainties from other sources (e.g., applied offset and bias corrections)
are propagated through to the final product, i.e. every ozone concentration has an associated uncertainty. The development of
the BSVertOzone database is described in detail in Hassler et al. (2018). For this study, the database was extended to cover the
period 1979 to 2019.

## 3 Ozone mass deficit

As in Bodeker and Scourfield (1995), the Antarctic vortex period (AVP; day 200-335; 19 July-1 December) mean ozone mass
deficit has been calculated for each year and is plotted together with an estimate of the EEASC in Fig. 1. The mass deficit
quantifies the mass of ozone that would need to be added to the atmosphere to return TCO values over Antarctica to above 220





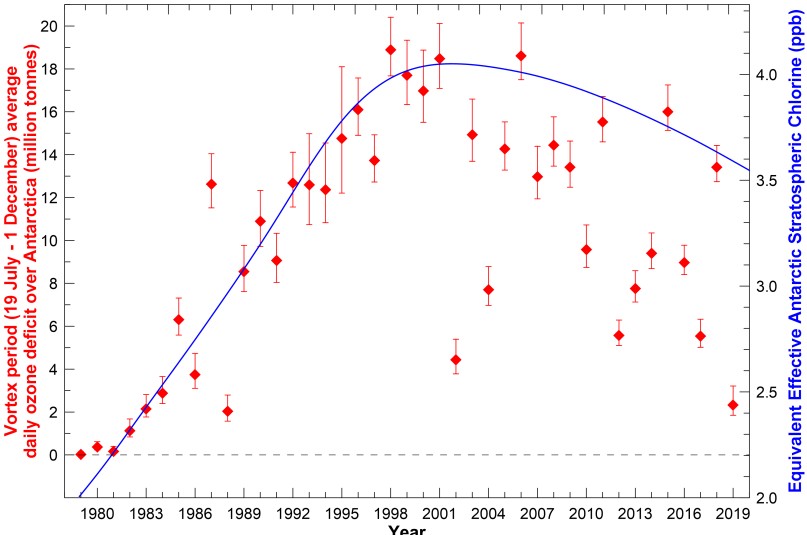

**Figure 1.** (red) Antarctic vortex period (AVP; day 200 to 335) average daily ozone mass deficit plotted against the left Y axis and (blue) Equivalent Effective Antarctic Stratospheric Chlorine (EEASC) plotted against the right Y axis.

DU (1 DU = $2.69 \times 10^{16}$ molecules/cm$^2$). The scale selected for the EEASC curve (right Y axis) maximizes the correlation between the AVP mean ozone mass deficit and EEASC over the period 1979 to 2000, just before EEASC peaks. The fact that after 2000 more of the data points fall below the EEASC curve than above it suggests that factors other than the decline in halogen loading of the Antarctic stratosphere is driving the return of the Antarctic ozone layer to pre-1980 levels.

The anomalously low AVP mean ozone mass deficits in 1988, 2002 and 2019 all result from sudden stratospheric warmings (SSWs), large in 1988 (Kanzawa and Kawaguchi, 1990), major in 2002 (Newman and Nash, 2005), and minor in 2019 (Wargan et al., 2020) that elevated Antarctic stratospheric temperatures and curtailed the heterogeneous chemical processes driving polar ozone destruction. The SSW in 1988 led to an Antarctic ozone hole that was shallow in depth and small in area (Kanzawa and Kawaguchi, 1990; Schoeberl et al., 1989; Krueger et al., 1989). In 2002, unusually large planetary wave activity caused a major

SSW that weakened and warmed the polar vortex, and resulted in reduced ozone depletion over Antarctica (Allen et al., 2003; Glatthor et al., 2004; Konopka et al., 2005; Manney et al., 2005; Ricaud et al., 2005). The minor SSW in September 2019 resulted in significantly higher than usual polar TCO (Wargan et al., 2020; Safieddine et al., 2020) and the much reduced AVP mean ozone mass deficit in that year, smaller than in 2002 ($4.43^{+0.96}_{-0.65} \times 10^9$ kg in 2002 cf. $2.32^{+0.89}_{-0.48} \times 10^9$ kg in 2019). The large difference in the severity of Antarctic ozone depletion in 2018 and 2019 was found by Wargan et al. (2020) to result from

(i) the geometry of the 2019 vortex, with ozone-rich middle-stratospheric air masses overlying the lower portion of the vortex, and (ii) significantly reduced vortex volume.

The smaller than expected ozone holes in 1988, 2002 and 2019 (as well as other years when the stratosphere was anomalously warm) should not be interpreted as the Antarctic ozone layer recovering from the effects of ODSs faster than expected (Safieddine et al., 2020).



## 4   Antarctic ozone hole area

Daily measures of the Antarctic ozone hole area, defined using the four criteria listed above, for the most recent four years (2016-2019) are shown in Fig. 2 in the context of the mean and maximum over the period 1979 to 2015. For two of the metrics

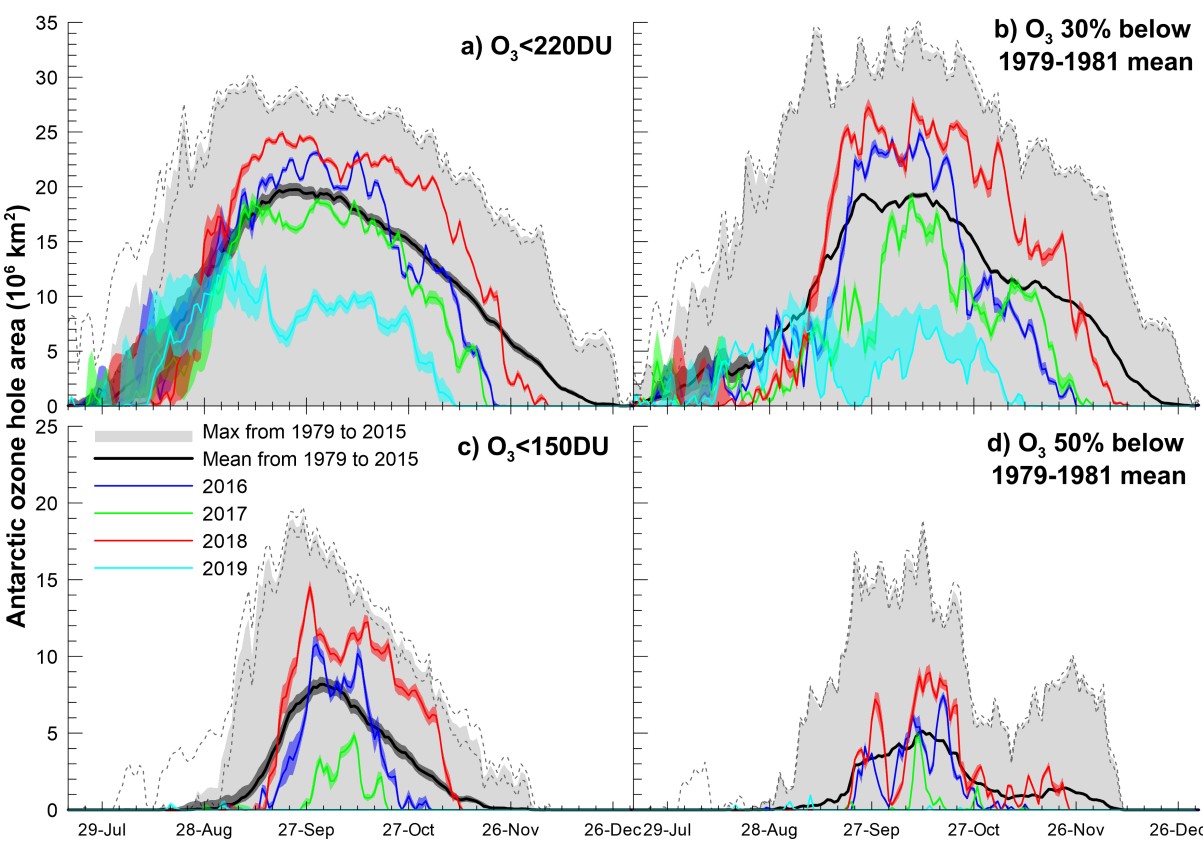

**Figure 2.** Daily measures of the size of the Antarctic ozone hole using four different criteria. Values for 2016 (blue), 2017 (green), 2018 (red) and 2019 (cyan) are compared against the range of values over the period 1979–2015 (greyed area). The mean ozone hole area over the period 1979 to 2015 is shown using a thick black line in all 4 panels. Coloured shaded regions around each trace indicate $\pm 1\sigma$ uncertainties.

(TCO less than 150 DU and TCO 50 % or more below the 1979–1981 climatology) the 2019 ozone hole is essentially absent. Early in the season, e.g. before the end of August, the uncertainties on the calculated ozone hole areas are larger than later in the season as a result of the large uncertainties in the filling of the TCO fields during polar darkness. Uncertainties are also larger when the TCO in a cell identified as an 'ozone hole' grid cell is very close to the threshold (such as in 2019 using the TCO 30 % or more below the 1979–1981 climatology threshold). A small reduction in TCO values can result in a large increase in the region identified as being depleted in ozone. Other than under these circumstances, the uncertainties on the ozone hole areas appear small, suggesting that the area metric is robust against uncertainties in the underlying TCO fields. The significant differences between the 2018 and 2019 Antarctic ozone holes have been discussed extensively by Wargan et al. (2020). The



source of the local minima in late October/early November in the time series for TCO 50 % or more below the 1979–1981 mean is discussed in Bodeker et al. (2005).

Annual maximum ozone hole areas, and the dates on which they occur, are shown for all four threshold conditions in Fig. 3. The annual maxima in the daily values of the Antarctic ozone hole areas for the four ozone hole area criteria peak around the

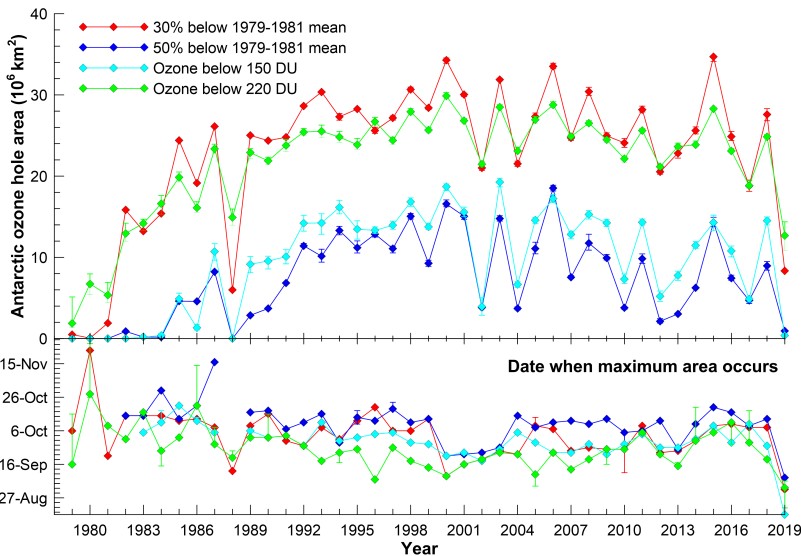

**Figure 3.** The annual maximum ozone hole area for the four different threshold criteria (upper panel), and the dates on which those maxima were achieved (lower panel). Error bars show $1\sigma$ uncertainties which are often asymmetrical.

turn of the century, close to when EEASC peaks (see Fig. 1). In contrast, the date when the maximum occurs shows a steady drift towards earlier dates over the 41-year period (linear trends equivalent to changes of between 15 and 19 days earlier, depending on metric). The cause of this drift towards consistently earlier dates of annual maximum ozone hole area has not been diagnosed here. The error bars, which are often asymmetric, are generally small.

The dates of disappearance of TCO values flagged as being within the Antarctic ozone hole by the four different criteria are

shown in Fig. 4. After drifting to later in the year over the first 1.5 decades of the data series, since the early to mid-1990s the dates of disappearance of ozone hole type values appear to have drifted to earlier in the year. As in Bodeker et al. (2005), we have considered to what extent earlier breakdown of the dynamical vortex in recent years may have contributed to the date of disappearance of ozone hole type values coming earlier in recent years.

To that end we have calculated 6-hourly profiles of the 550 K meridional impermeability ($\kappa$) against equivalent latitude

(Bodeker et al., 2002) from the NCEP-CFSv2 (the National Centers for Environmental Prediction Climate Forecast System Version 2, Saha et al., 2014) reanalyses for the period 1979 to 2019. For each year, 1460 meridional maximum $\kappa$ values were extracted (noting the 6-hourly frequency of the reanalyses), hereafter $\kappa_{max-day}$. The date each year when $\kappa_{max-day}$ falls below 20 % of the $95^{th}$ percentile of $\kappa_{max-day}$ is then identified as the date of the breakdown of the dynamical vortex. The annual dates of disappearance of ozone hole type values for the four different criteria are plotted against the dynamical





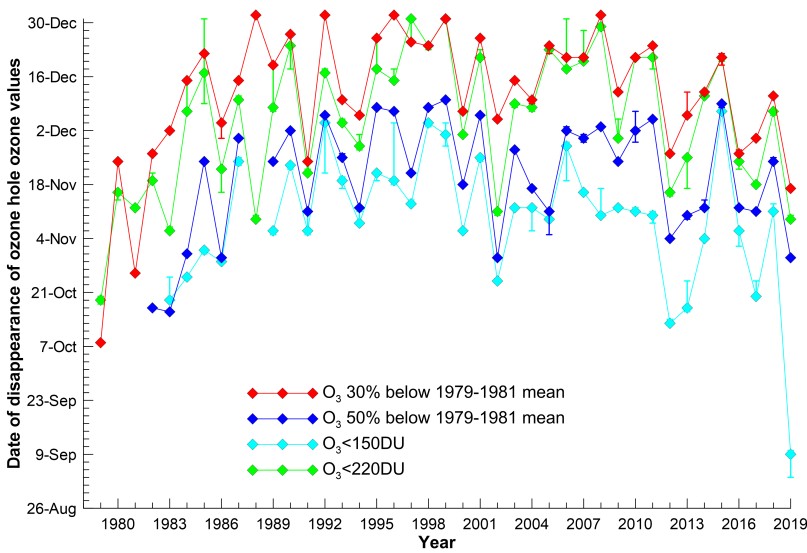

**Figure 4.** The annual dates of disappearance of ozone hole type values for all four threshold criteria. In some years the threshold for being identified as an ozone hole type value is not reached.

vortex breakdown dates in Fig. 5. Except for the criterion of TCO being 30 % or more below the 1979 to 1981 mean, the date of the dynamical breakdown of the Antarctic polar vortex explains more than half of the variance in the dates when ozone hole type values disappear. Furthermore, for 2002 and 2019 when a major SSW and a minor stratospheric warming occurred, respectively, both the dates on which ozone hole type values disappeared and the dates when the dynamical vortex broke down, were earlier than in most other years (see labels on Fig. 5).

**5  Polar cap means**

While some previous studies have reported on annual minimum TCO values within the Antarctic vortex as a metric for tracking Antarctic ozone depletion (including Bodeker et al., 2005), Müller et al. (2008) showed that the utility of examining the minimum in daily TCO poleward of a threshold latitude was debatable, insofar as it relies on a single measurement. Müller et al. (2008) found that, for Arctic conditions, the minimum value often occurs in air *outside* the polar vortex, both in the

observations and in a chemistry-climate model and that the minimum value does not show a good correlation with chemical ozone loss in the vortex deduced from observations. They recommended that the minima, relying on a single measurement, should not be used as a metric of polar ozone depletion. Following that recommendation, we consider rather daily TCO zonal means calculated against equivalent latitude on the 550 K surface (Bodeker et al., 2001b). Examples of such zonal mean TCO profiles by equivalent latitude for 1 October of each year are shown in Fig. 6. The meridional profiles by equivalent latitude

are characterised by very steep gradients across the dynamical polar vortex edge, typically around 62° S equivalent latitude (Bodeker et al., 2002), and very shallow gradients through the core of the dynamical vortex poleward of 75°S equivalent





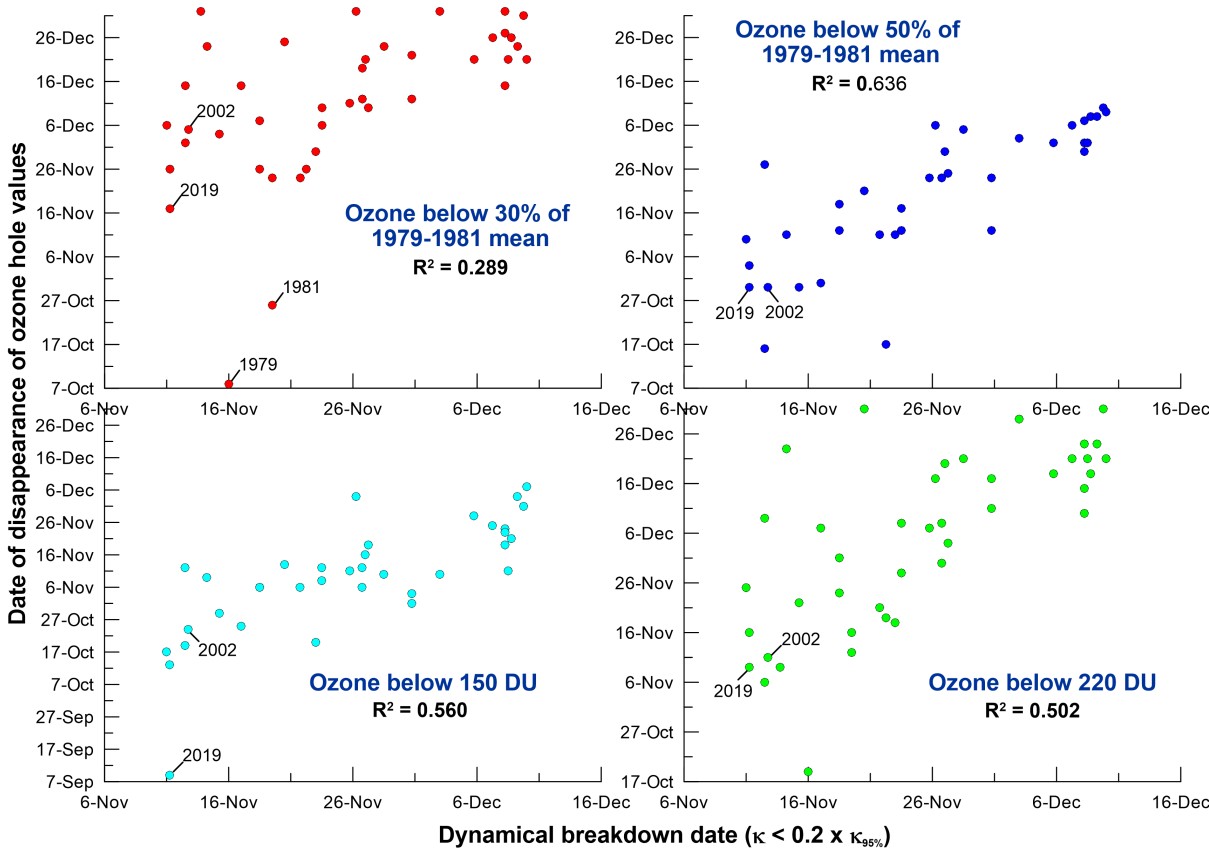

**Figure 5.** The annual dates of disappearance of ozone hole type values for all four threshold criteria plotted against the annual date of the breakdown of the dynamical vortex. $R^2$ values for linear fits to the data plotted in each panel are also shown.

latitude. It is for this reason that 'polar cap' TCO means are calculated in this region (see below). Some years, such as 2002, 2004 and 2019 show meridional profiles of zonal means much higher than would be expected from the EEASC loading at that time for the reasons outlined above.

Polar cap means (mean of TCO poleward of 75°S equivalent latitude) have been calculated for each day and the daily time series for 2016 to 2019 are plotted in the context of the range and mean from 1979 to 2015 in Fig. 7. Uncertainties on the polar cap means are larger during the winter period where the uncertainties on the filled fields are larger. By mid to late August, the uncertainties on the source TCO fields have only a very small effect on the uncertainties on the calculated polar cap means. The effects of the minor stratospheric warming that occurred in September 2019 are clear in the elevated polar cap means during

that period. In mid-October, the 2018 polar caps means came close to being record low values for this time of the year.

    Annual minima in the daily polar cap mean TCO time series, and the dates on which those minima occurred, are shown in Fig. 8. The values minimize around the turn of the century and show a small positive trend thereafter. The uncertainties on the annual minimum polar cap mean TCO are typically less than 5 DU from 1979 to 1995 and less than 2 DU thereafter.





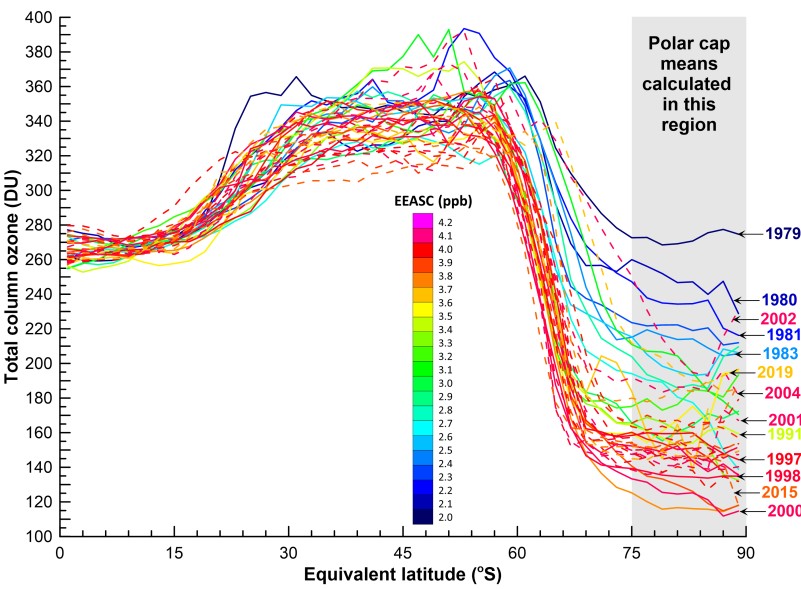

**Figure 6.** Zonal mean TCO profiles by equivalent latitude calculated on the 550 K surface for 1 October of each year. Each trace is coloured by Equivalent Effective Antarctic Stratospheric Chlorine (EEASC) shown in the colour-scale insert in the figure. Solid lines show data for the period 1979-2000 while dashed lines show data for the period thereafter. Uncertainties have been excluded from these traces for clarity but are generally very small (i.e. less than 5 DU at this time of the year).

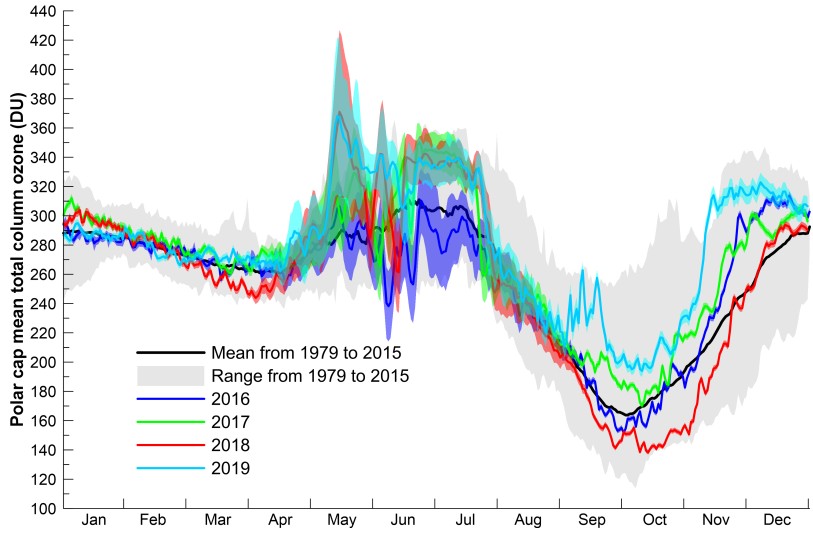

**Figure 7.** Daily polar cap average (75°S to 90°S equivalent latitude) total column ozone. The last four years of data are shown in the context of the 1979 to 2015 climatology. The shading around each trace indicates the $1\sigma$ uncertainties.





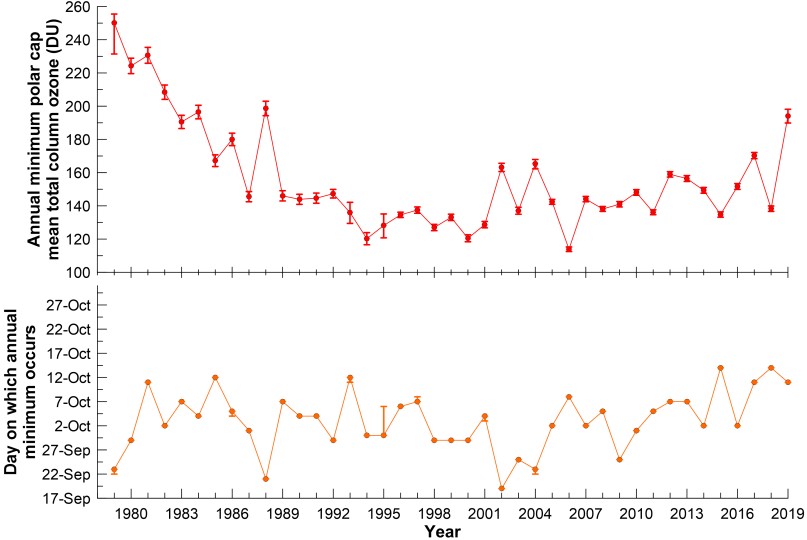

**Figure 8.** Annual minima in the daily polar cap mean TCO time series and the dates on which those minima occurred. Error bars show the $\pm 1\sigma$ uncertainties which are often asymmetrical.

## 6  Changes in the vertical distribution of ozone

Using vertically resolved ozone concentration measurements obtained from the BSVertOzone database, partial columns in 1 km thick layers poleward of 75°S equivalent latitude were calculated for the period 1985 to 2019. Partial columns of 1 km vertical extent, centred on 12 to 21 km altitude, are shown for the Southern Hemisphere spring in Fig. 9. Again, the anomalous warm winters of 1988, 2002 and 2019 resulting in more ozone are clearly visible. Across all spring months partial ozone columns have increased since the late 1990s. The percentage contribution of each 1 km thick layer to the monthly mean, polar

cap mean partial ozone column between 11.5 and 21.5 km is shown in Fig. 10. It is not clear whether the significant shifts in ozone between layers in September in the mid-1990s result from sampling biases in the measurements available (noting the screening of SAGE-II data below 23 km in the few years following the Mt. Pinatubo volcanic eruption in June 1991) or whether the vertical redistribution reflects a physical response to the eruption. During October and November the general sense is that from 1985 to around the turn of the century, ozone in the 11.5 and 21.5 km column is concentrated more in the upper part of

the column (20 to 21 km) and less in the lower part of the column (13 to 18 km) as a result of the heterogeneous chemistry in the Antarctic being most active between 16 and 18 km (Hofmann et al., 1987; Johnson et al., 1992). This trend reverses after the turn of the century with ozone showing a more equitable distribution across the 10 layers by 2019.





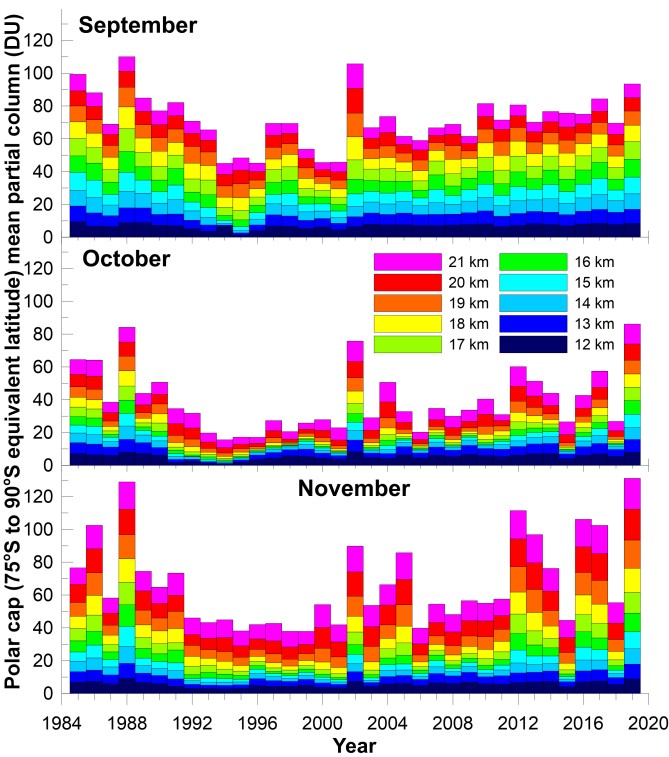

**Figure 9.** Monthly mean, polar cap mean (75° S to 90° S equivalent latitude), 1 km thick partial ozone columns for September, October and November (Southern Hemisphere spring) of each year from 1985 to 2019.

## 7 Conclusions

Several metrics describing the evolution of the Antarctic ozone hole over the 41-year period 1979 to 2019 are reported on above.
These analyses were only possible through the availability of a complete, homogeneous climate data record of daily TCO fields. As detailed in Bodeker et al. (2020a), significant effort is required to homogenize the ozone fields from the 17 different space-based sensors measuring ozone that comprise the BS–filled TCO database, as well as to infer missing data through the polar night and in other regions where the operational parameters of the satellites result in data gaps. The requirements of the GCOS (Global Climate Observing System; GCOS-138, 2010; Bojinski et al., 2014) for climate data records, and in particular the need
for all data to have traceable uncertainties, has led to the most recent versions of the NIWA–BS and BS–filled TCO databases (V3.4 and V3.5.1) including estimates of the uncertainties on every TCO value as described in Bodeker et al. (2020a). This has allowed, for the first time, uncertainties to be included on the Antarctic ozone depletion metrics, showing which metrics are sensitive to uncertainties in the source TCO fields and which are not.

While a formal attribution of changes in the metrics shown above to changes in Antarctic stratospheric halogen loading has
not been made and, as a result, statements about the recovery of the Antarctic ozone layer from the effects of ODSs cannot be made, all of the metrics directly related to ozone levels over Antarctica, i.e. AVP mean depleted mass, annual maximum





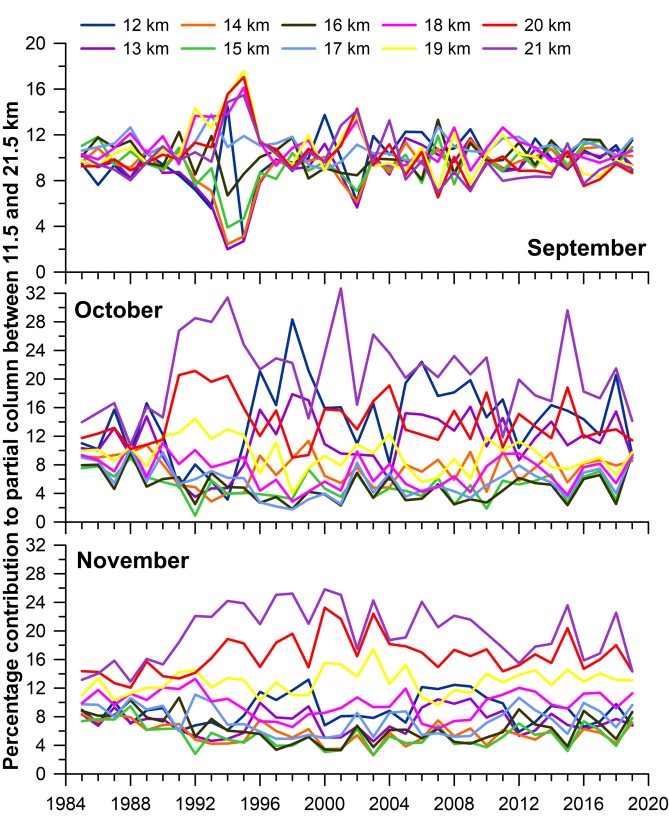

**Figure 10.** Percentage contribution of each 1 km thick layer to the monthly mean, polar cap mean (75° S to 90° S equivalent latitude), partial ozone column between 11.5 and 21.5 km for September, October and November (Southern Hemisphere spring) of each year from 1985 to 2019.

ozone hole areas under four different criteria, the date when ozone hole type values disappear, annual minimum polar cap mean TCO, and polar cap mean ozone partial columns between 11.5 and 21.5 km, all show changes consistent with decreasing Antarctic stratospheric halogen loading. It is important that the network of ground-based and space-based instruments required

to monitor the global ozone layer are maintained, that global homogenised and quality-controlled TCO climate data records continue to be maintained, and that metrics of Antarctic ozone depletion continue to be updated so that the effectiveness of the Montreal Protocol and its Amendments and Adjustments in returning the Antarctic ozone layer to an unperturbed state can continue to be assessed by policymakers.

*Data availability.*  The NIWA–BS TCO database (doi:10.5281/zenodo.1346424, Bodeker et al. (2018)) and the BS–Filled TCO database

(doi:10.5281/zenodo.3908787, Bodeker et al. (2020b)) are available from http://www.bodekerscientific.com/data/total-column-ozone and from the zenodo archive. Both databases are available for non-commercial purposes under the Creative Commons Attribution Non Commer-





cial Share Alike 4.0 International licence and are provided as netCDF files. The data underlying all of the plots in this paper can be obtained on emailing the lead author.

*Author contributions.* GEB wrote much of the code for processing the TCO data files into a common format, the code for creating the
BS–filled TCO database, created several of the figures and wrote much of the text of the paper. SK ran and debugged the code for applying the corrections to the 17 different TCO data sets, assisted with the writing of the paper, and created some of the figures. SK also created the vertically resolved ozone database used to create Fig. 9 and Fig. 10.

*Competing interests.* The authors declare no competing interest.

*Acknowledgements.* The analyses presented above and the writing of this paper was supported through funding from the New Zealand Min-
istry of Business, Innovation and Employment through the Antarctic Science Platform (ASP) contract ID ANTA1801. We acknowledge the World Meteorological Organization-Global Atmosphere Watch (WMO/GAW) Ozone Monitoring Community, World Ozone and Ultraviolet Radiation Data Centre (WOUDC) for the Dobson and Brewer TCO data which were used to correct the space-based measurements used in this analysis. A list of all contributing sites is available on https://search.datacite.org/works/10.14287/10000001. We would also like to thank the many satellite teams who provide their data freely to the ozone research community. Without access to these source data, the calculation
of the Antarctic ozone depletion metrics presented here would not be possible.



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
