# Peer review of "Indicators of Antarctic ozone depletion: 1979 to 2019"

_Atmospheric Chemistry and Physics, 2020_

## Referee Comment (RC1) · Anonymous Referee #1 · 3 Dec 2020

**1   Overall Remarks**

This is a comprehensive and well written paper on the long-term evolution of the Antarctic ozone hole using a number of measures. It is well suited for publication in ACP, and I have only a few minor suggestions. If at all possible, it would be great to also include the 2020 ozone hole, which was rather on the large size again.

**2   Comments**

line 4: "Antarctic" should be "Antarctica", or "the Antarctic"

line 28: replace "has led" by "should lead". Even in 2020, we are still far from "recovered".

line 56: delete "a continuous body of"?

line 91: The correlation between ozone mass deficit and EEASC is the same, no matter how you scale the axes. What you did, is obtain the axis scaling from a linear fit of EEASC to mass deficit. So: delete "selected", replace "maximizes ... between" by "is obtained from fitting", replace "deficit and EEASC" by "deficit to EEASC"

lines 93, 94: Yes, it is really interesting that the data points scatter so much more after 2000, compared to before. Did Southern Hemisphere meteorology become much more variable? Does that have anything to do with the shift of climate patterns / jet streams in the Southern Hemisphere due to climate change and the ozone hole? You might want to add some text and cite a few references (e.g. recent ozone assessments)?

Figs. 1, 3, 4, 8: it might be useful to show / mention the correlation of the different time series with EEASC (overall correlation over the entire time series). Is there one measure that correlates best with EEASC? That might be the best measure to capture a dependence of the Antarctic ozone hole on ozone depleting substance loading.

A well written paper! It is very rare that I have so few comments on a paper.

---

## Author Comment (AC1) · 19 Feb 2021

**Overall Remarks**

This is a comprehensive and well written paper on the long-term evolution of the Antarctic ozone hole using a number of measures. It is well suited for publication in ACP, and I have only a few minor suggestions.

We thank the reviewer for taking the time to review the paper.

If at all possible, it would be great to also include the 2020 ozone hole, which was rather on the large size again.

We appreciate the reviewers comment but, unfortunately, this won't be possible. The analysis reported on in the paper is based on the BS-filled database that extends from 1979 to 2019. Updating the database by just a single year is a significant amount of work that we are not currently resourced for. There will always be a desire to have 'just one more year added'.

**Comments**

line 4: "Antarctic" should be "Antarctica", or "the Antarctic"

Thanks for catching that. Now changed to 'Antarctica'.

line 28: replace "has led" by "should lead". Even in 2020, we are still far from "recovered".

Agreed. We have now changed this to 'is expected to lead'.

line 56: delete "a continuous body of"?

Agreed and replaced 'a continuous body of' with 'extant'.

line 91: The correlation between ozone mass deficit and EEASC is the same, no matter how you scale the axes. What you did, is obtain the axis scaling from a linear fit of EEASC to mass deficit. So: delete "selected", replace "maximizes ... between" by "is obtained from fitting", replace "deficit and EEASC" by "deficit to EEASC"

Very good point. Thanks and we have made the suggested changes.

lines 93, 94: Yes, it is really interesting that the data points scatter so much more after 2000, compared to before. Did Southern Hemisphere meteorology become much more variable? Does that have anything to do with the shift of climate patterns / jet streams in the Southern Hemisphere due to climate change and the ozone hole? You might want to add some text and cite a few references (e.g. recent ozone assessments)?

We have now added some text and citations to provide context for the observation of anomalously smaller AVP mean ozone mass deficit after 2000.

Figs. 1, 3, 4, 8: it might be useful to show / mention the correlation of the different time series with EEASC (overall correlation over the entire time series). Is there one measure that correlates best with EEASC? That might be the best measure to capture a dependence of the Antarctic ozone hole on ozone depleting substance loading.

Excellent idea and now done. Ozone hole area below 220 DU has the highest correlation against EEASC. Additional commentary has been added to each of those figures and to the conclusion of the paper to that effect.

A well written paper! It is very rare that I have so few comments on a paper

Thank you again for the feedback.

---

## Author Comment (AC2) · 19 Feb 2021

We thank reviewer #2 for taking the time to review the paper.

Review of "Indicators of Antarctic ozone depletion: 1979 to 2019"
This is an extremely clear, well written manuscript on which I have only a small number of comments. The authors provide a clear update on Antarctic ozone depletion through a variety of established metrics, benefiting from recent advancements in the filling of the total column ozone database. My main comment relates to the handling and production of uncertainties (paragraph beginning L69). In my interpretation, you perturb the TCO field by adding/subtracting the gridded uncertainty values.

[Figure]

That is correct.

Let TCO values be x=x1,. . .xn, the corresponding uncertainty be $\sigma=\sigma 1,. . .\sigma n$, and $\alpha$ be a function for a metric, for example for a uniform area average $\alpha=1/n,. . .1/n$ n times. Your metric value for the three databases (added uncertainty, unperturbed, subtracted uncertainty) are then $\sum_n^i (x_i + \sigma_i)\alpha_i, \sum_n^i x_i\alpha_i$, and $\sum_n^i (x_i - \sigma_i)\alpha_i$. If I have understood correctly, the uncertainty for the metric is found by calculating the difference between the perturbed and unperturbed metric values found in the previous sentence. Therefore, you are estimating uncertainty as $\sum_n^i \sigma_i\alpha_i$. If I have understood the above correctly, I don't see why this uncertainty estimate is an appropriate one as it doesn't consider that gridded uncertainty estimates will be correlated.

Whether or not the uncertainty estimates are correlated is more a function of the construction of the underlying TCO database rather than in how the uncertainties are incorporated into the net uncertainty on each metric. In calculating the net uncertainty on each metric we assume that the uncertainties are perfectly correlated in that they are *all* -1$\sigma$ or *all* +1$\sigma$. We do this to create conservative (i.e. worst case) uncertainties. If we assume the uncertainties to be uncorrelated, then the resultant uncertainties on the metrics tend to be unrealistically small.

I am not aware of other uses of uncertainty estimation like this and given that a lot of figures and results from this manuscript rely on this method of uncertainty estimation I think there should be some explanation or references as to its efficacy.

We have added text to the paper to better explain this choice of methodology.

Minor comments: L4: Antarctic → Antarctica

Thanks for catching that. Now changed to 'Antarctica'.

L44: I think there should be an 'and' between temperatures and drives.

No, the way we had it was correct, i.e. 'Interannual variability in Antarctic stratospheric dynamics...drives significant interannual variability in the severity of Antarctic ozone depletion. Has we used the word 'manifests' rather than 'manifest', then you would be correct, though we would then needed to have removed the parenthetical commas.

L91: Correlation is unchanged by the Y axes scale choices, so how was the Y axis scale chosen? Presumably to achieve the best fit between EEASC and ozone deficit pre 2000.

Yes, and we have now clarified that in the text.

Fig2: I assumed that the dashed grey lines are the uncertainty of the max area values,but it isn't mentioned.

Good point and now added.

Fig7: May I suggest that the opacity of the coloured shading be decreased for clarity?

Sure and fill colour opacity has been reduced from 50% to 35% in all four plots.